# Pre-Clinical Cell Therapeutic Approaches for Repair of Volumetric Muscle Loss

**DOI:** 10.3390/bioengineering7030097

**Published:** 2020-08-20

**Authors:** Mahdis Shayan, Ngan F. Huang

**Affiliations:** 1Department of Cardiothoracic Surgery, Stanford University, Stanford, CA 94305, USA; mahdis.shayan@stanford.edu; 2Stanford Cardiovascular Institute, Stanford University, Stanford, CA 94305, USA; 3Veterans Affairs Palo Alto Health Care System, Palo Alto, CA 94304, USA

**Keywords:** tissue engineering, cell transplantation, stem cells, satellite cells, induced pluripotent stem cells, volumetric muscle loss

## Abstract

Extensive damage to skeletal muscle tissue due to volumetric muscle loss (VML) is beyond the inherent regenerative capacity of the body, and results in permanent functional debilitation. Current clinical treatments fail to fully restore native muscle function. Recently, cell-based therapies have emerged as a promising approach to promote skeletal muscle regeneration following injury and/or disease. Stem cell populations, such as muscle stem cells, mesenchymal stem cells and induced pluripotent stem cells (iPSCs), have shown a promising capacity for muscle differentiation. Support cells, such as endothelial cells, nerve cells or immune cells, play a pivotal role in providing paracrine signaling cues for myogenesis, along with modulating the processes of inflammation, angiogenesis and innervation. The efficacy of cell therapies relies on the provision of instructive microenvironmental cues and appropriate intercellular interactions. This review describes the recent developments of cell-based therapies for the treatment of VML, with a focus on preclinical testing and future trends in the field.

## 1. Introduction

Volumetric muscle loss (VML) is the traumatic, surgical or degenerative loss of a substantial portion of bulk skeletal muscle in a manner that overwhelms the endogenous repair capacity of the muscle and results in impaired scar tissue formation. VML is associated with chronic functional disability in many military and civilian populations following battlefield injuries, car accidents, tumor ablation or degenerative diseases [1,2]. VML lacks a standard treatment to replace the lost tissue with contractile muscle or restore muscle strength. Current surgical approaches, such as autologous free flap grafting, scar tissue debridement or minced skeletal tissue transfer, are utilized to reconstruct the tissue defects [3,4]. However, these techniques are limited by tissue availability, the need for highly skilled surgeons, significant donor site morbidity and functional deficiencies. Furthermore, the efficacy of these techniques for the repair of severe injuries, such as VML, has not been demonstrated clinically.

Skeletal muscle consists of bundles of oriented multi-nucleated muscle fibers (myofibers) that are surrounded by connective tissue, along with branches of blood vessels and peripheral nerves that supply blood flow and electrical signals to the muscle. Thus, a variety of cell types work in concert to form the skeletal muscle structure (Figure 1). Following injury, skeletal muscle normally regenerates new myofibers after the inflammatory and repair phases of healing, leading to restored muscle structure and function. However, in the case of traumatic muscle injury that results in VML, chronic inflammatory responses persist, and fibrosis and scar formation override muscle regeneration. Significant loss of satellite cells and native muscle extracellular matrix (ECM), as well as fibrosis and chronic inflammation, prevent muscle regeneration and cause functional deterioration of the damaged muscle. Traditional rehabilitative or physical therapy alone has not been effective in restoring the strength and function of muscle tissue in VML injuries, because the loss of muscle fibers is so significant. Recently, however, a 13-patient cohort study that combined physical therapy with the use of scaffolds derived from decellularized porcine urinary bladder reported that 6 months after scaffold implantation muscle, strength and range of motion significantly improved in patients, by an average of 37.3% and 27.1%, respectively [5]. Although using a combination of biological scaffolds and physical therapy is promising, because of the polytraumatic nature of VML injuries, which include neural, vascular and tendon damage as well as bone fractures, and the presence of extensive fibrosis within and surrounding the muscle tissue, the need for an effective treatment that will regenerate the lost muscle fibers as well as other types of injured tissues de novo and restore muscle function is in high demand. Cell transplantation and tissue engineering approaches represent a promising solution for restoring muscle function. Some of the key considerations in tissue engineering approaches include the choice of therapeutic cell type, cell delivery method, and host immune response to the transplanted cells. In this review, we summarize the current state of cellular therapies for VML by reviewing several cells which are present in skeletal muscle tissue during normal and pathological conditions. We explore multiple potential cell sources that could play a role in muscle regeneration for VML repair, with a focus on cellular therapies at the preclinical stage. 

## 2. Main Myogenic Cell Sources 

A variety of cells reside within skeletal muscle, including muscle satellite cells, pericytes, vascular lineages, interstitial stem cells and fibro/adipogenic progenitors (FAPs). Among these cell types, satellite cells are primarily responsible for natural skeletal muscle repair and regeneration. Satellite cells are muscle stem cells (MuSCs) that reside beneath the basal lamina surrounding each myofiber. These cells are a heterogenous population, but commonly express paired box protein-7 (Pax7), a transcription factor that plays role in myogenesis and regulates the proliferation of MuSCs. In response to myofiber injury, quiescent satellite cells activate, proliferate, and give rise to myogenic progenitor cells (MPCs). MPCs differentiate into myoblasts and can fuse to form multinucleated myotubes, which mature into newly formed myofibers. 

Although MuSCs are considered promising cell sources for VML treatment, some of the limitations associated with satellite cells include their low abundance within muscle, challenges in isolating and purifying these cells, limited self-renewal and differentiation potential in vitro, and low engraftment post-transplantation [6,7]. The in vitro expansion of satellite cells can result in the loss of their innate myogenic behavior, but multiple research studies have demonstrated that the regenerative capacity of satellite cells can be regulated by biochemical and biophysical microenvironmental cues, such as peptide functionalization, substrate stiffness and three-dimensionality [8,9,10,11]. For example, Gilbert et al. showed that hydrogels of muscle-like stiffness could recapitulate the rigidity features of the stem cell niche. MuSCs cultured on these 12 kPa stiffness hydrogels could preserve self-renewal characteristics in vitro, in contrast to commonly used rigid plastic culture dishes [11]. Additionally, Pruller et al. showed that murine MuSCs embedded in collagen I, polyethylene glycol (PEG)-fibrinogen and three-dimensional (3D) fibrin scaffolds did not show any myogenic differentiation [10]. However, placing freshly isolated myofibers within 3D scaffolds significantly improved the myogenic behavior of satellite cells on the myofibers, in part because the cells were still in their native niche [10].

Given the limitations of MuSCs, mesenchymal stem cells (MSCs) are an alternative cell source. The intramuscular injection of MSCs has accelerated muscle repair in injured skeletal muscle in animal models [12]. MSCs are a class of non-hematopoietic multipotent adult stem cells which can differentiate into many cell types, including osteoblasts, adipocytes, chondrocytes and myoblasts. They are present in a variety of adult tissues, including bone marrow, birth-derived tissues and dental pulp, however bone marrow and adipose tissue are the two main sources of MSCs. Compared to satellite cells, MSCs have greater abundance, and can be harvested by minimally invasive processes like bone marrow extraction or lipoaspiration [13]. MSCs are immunologically advantageous because of their low expression level of major histocompatibility complex class I (MHC I) and II (MHC II) proteins, which makes them attractive for transplantation [14]. Although MSCs can be a suitable cell source for clinical treatment of VML in future, many studies have shown the low engraftment of MSCs and a lack of tissue-specific differentiation. It is not clear that MSCs are beneficial in muscle regeneration just by secreting immunomodulatory and paracrine signals, or by myogenic differentiation. More research studies should be conducted to optimize MSC isolation and the myogenic differentiation process. The identification of suitable microenvironmental factors that support the myogenic phenotype will be important for harnessing the therapeutic potential of MSCs, as is the understanding of how MSCs participate in muscle regeneration either by paracrine signals or myogenic differentiation. 

Induced pluripotent stem cells (iPSCs) represent another attractive myogenic cell source for skeletal muscle regeneration, due to their unlimited self-renewal capacity and capacity to differentiate into myogenic cells. The iPSCs can be directly reprogrammed into skeletal muscle cells via the overexpression of myogenic transcription factors, such as Pax7 or MyoD, or by small molecules [15,16]. Genetic reprogramming is more efficient than small molecule-based reprogramming, and can generate 3D contractile skeletal muscle that fuses with the existing damaged tissues [17]. However, the risks of genetic instability and tumorgenicity make reprogramming unreliable for clinical studies and VML treatment. Besides these main cell types, other potential cell sources for skeletal muscle regeneration are summarized in Table 1, and have been broadly discussed elsewhere [18]. Few studies have specifically investigated many of these myogenic cell sources, such as pericytes or mesoangioblasts, for VML repair. The critical factors for choosing a suitable myogenic cell source for treatment of VML include the potential for myogenic differentiation, availability, ease of isolation and sorting, as well as ease of in vitro expansion and immunogenicity. Adipose-derived MSCs (ASCs) are abundant and easy to isolate from adipose tissue with low immunogenicity. However, it is not clear that they directly differentiate into muscle cells, or whether their impact is more immunomodulatory using paracrine signals. In Table 1, different myogenic cell sources are compared, and their pros and cons are outlined. 

## 3. Non-Myogenic Cells

In addition to resident MuSCs, the normal functioning and regeneration of skeletal muscle tissue are supported by fibroblasts, endothelial cells, neural cells, fibro/adipogenic progenitor (FAP) cells and infiltrating immune cells. FAPs are muscle-resident progenitor cells of mesenchymal origin, expressing stem cells antigen-1 (Sca-1) and platelet-derived growth factor receptor beta (PDGFR-β) surface markers, which can differentiate into fibroblasts and adipocytes. Following muscle injury, FAPs proliferate and enhance myogenic differentiation by generating pro-differentiation signals for MPCs [25]. The intercellular interactions and paracrine effects of the above-mentioned support cells on MuSCs are important for the regeneration of damaged nervous and vascular tissues that are often associated with VML [26,27]. For example, pericytes and MSCs regulate myogenesis through paracrine effects on other cell types, such as macrophages. Chronic inflammation and the delayed transition of inflammatory to anti-inflammatory phase in VML impairs the hemostasis of many intracellular interactions, and results in fibrotic muscle degeneration. Persistent inflammatory signals such as TGF-β prevent FAPs, apoptosis and lead to their pathological accumulation and differentiation into fibroblasts [28]. MSCs derived from bone marrow or adipose tissue can ameliorate the local immunological response by suppressing inflammatory cytokines and modulating local immune cells in the injured muscle, as discussed in [29].

Multi-cellular culture systems have been developed to study the intercellular and paracrine interactions that are necessary for muscle formation and function. For example, Ostrovidov et al. demonstrated that the co-culture of a PC12 neural cell line with a C2C12 myoblast cell line could induce C2C12 myoblast differentiation and myotube formation [27]. Neural cell sources, such as human neural stem cells and iPSC-derived neurons, in co-culture with skeletal muscle cells could form functional neuromuscular junctions (NMJ) and have improved muscle differentiation and myotube formation [30]. Laternser et al. demonstrated that the interaction between tendon and muscle cells affects the functionality of the regenerated muscle. 3D muscle and tendon tissues were developed by printing human myoblasts and rat tenocytes within bioink layers. The printed myofibers were functional and could contract following electrical stimulation. The increased expression of muscle and tendon marker genes indicated the good differentiation of cells [31]. Additionally, endothelial cells modulate skeletal MuSCs differentiation and proliferation by secreting growth factors such as vascular endothelial growth factor (VEGF), insulin growth factor (IGF-1) and hepatocyte growth factor (HGF) [32]. These examples highlight the importance of both MuSCs and other supporting cells in mediating myogenesis.

## 4. Preclinical VML Treatment Studies

### 4.1. Animal Models of VML

Approximately 90% of preclinical models of VML have been conducted in mice and rats [33]. However, a standard VML model with respect to muscle anatomical location and defect size has not been defined. The latissimus dorsi (LD), tibialis anterior (TA), quadriceps and abdominal wall muscles are the most frequently ablated muscles in experimentally induced VML [33]. Although in most VML models more than 20% of the muscle mass is ablated, Anderson et al. reported that 15% muscle ablation was the critical threshold for irreversible muscle loss. The critical threshold was characterized by persistent fibrotic and inflammatory response, as well as incomplete innervation and partial myofiber regeneration [34]. Full-thickness injuries of 2, 3 and 4 mm diameters were created in the quadricep muscles of mice, which resulted in 5%, 15% and 30% muscle mass defects, respectively. In addition to the size of defect, other factors could mediate the healing outcome of the VML lesion; including whether the defect is partial or full-thickness, how the suture for wound closure was applied (e.g., bridging vs. non-bridging lesion) and the proportion of defect size to animal weight. In addition, myofiber density and the amount of satellite cells in the muscle tissue varies between male and female animals, and across different ages. Biomechanical loading is also different in each muscle tissue. Considering these parameters, VML rodent models (Table 2) are beneficial for examining therapies or understanding the pathophysiology of VML and skeletal muscle regeneration. However, the treatment results of rodent preclinical studies should not be extrapolated for human VML treatments without considering factors such as mechanical loading. In other words, the current rodent VML models fit better for small non-appendicular muscles like facial muscles than large, appendicular muscles.

### 4.2. Cell-Seeded Scaffolds for Preclinical Treatment of VML

The direct transplantation of MuSCs and MPCs into damaged tissue results in poor cell survival and limited engraftment due to a lack of sufficient cell support and the harsh injured tissue microenvironment [35]. Scaffolds support cell attachment, survival and differentiation, and also facilitate topographical, biomechanical and biochemical cues to promote myogenesis [33,36]. The transplantation of muscle stem cells seeded in decellularized bladder matrices into rat TA muscle with VML injury showed that muscle peak isometric torque was already enhanced by the use of scaffolds alone, but the inclusion of cells with the scaffolds doubled the functional recovery of the muscle [37]. Aligned nanofibrillar collagen scaffolds direct the alignment of murine myoblasts, and promote myotube fusion and contractility in the engineered muscle tissue [38]. Scaffolds, in addition to physically filling in the lost muscle volume and augmenting force transmission across the defect, can be used as combined delivery vehicles for cells and growth factors. Macroporous alginate hydrogels simultaneously delivered murine myoblasts along with a VEGF/IGF-1 cocktail. The results showed a significant increase in the muscle contractile force and a decrease in fibrosis [39]. A more extensive discussion of scaffold biomaterials for skeletal muscle regeneration in VML is found in [19,40].

### 4.3. MuSC-Based Therapies for Preclinical Treatment of VML

MuSCs derived from newborn mice and embedded in fibrin hydrogel were able to engraft and differentiate into new myofibers in NOD SCID mice with VML (Table 2). This led to a 30% increase in muscle mass and a 50% reduction in fibrosis area, compared to untreated muscles. The detection of LacZ^+^/Pax7^+^ in the cross section of engrafted muscle demonstrated that the transplanted MuSCs, labeled with LacZ, could restore satellite cells. In addition, LacZ^+^ donor-derived cells contributed to an average of 26% of the new vessels formed in engrafted regions [41]. Although this study showed evidence of neotissue formation in a VML animal model using histology and immunohistochemical analysis, no functional measurements were taken to compare recovered muscle with uninjured tissue, such as the level of nerve and force restoration, muscle strength deficits, and measurements of peak isometric tetanic force or torque.

The importance of intercellular interactions for the treatment of VML was examined by Quarta et al. Human MuSCs and four MuSC support cell populations (endothelial cells, FAPs, hematopoietic cells, fibroblast-like cells) were encapsulated in ECM-based hydrogels and transplanted into the injured TA muscle of NOD SCID mice (Table 2). The results show that support cell presence significantly enhances the viability of MuSCs, increases vascularization and reduces the fibrotic response by 50%. In addition, muscle mass in the presence of both support cells and MuSCs could be restored to 90% of the uninjured tissue mass, while bioconstructs with MuSCs alone could only recover up to 70% of mass. The in vivo force restoration of the transplanted cellular bioconstruct was 65% of the intact muscle, while NMJ formation remained significantly lower, yet could improve with exercise [42]. Additionally, we previously demonstrated the importance of endothelial interactions and spatially patterned scaffold topography in modulating muscle formation and revascularization in a mouse VML model. Recently, we further demonstrated that co-seeding C2C12 myoblasts with endothelial cells within anisotropic nanofibrillar collagen scaffolds resulted in the formation of endothelialized and highly organized myotubes, with increased cell survival in vivo, higher secretions of angiogenic cytokines and more synchronized contractility compared to endothelialized myotubes formed on randomly-oriented scaffolds. Similar results in forming aligned myotubes and enhancing vascular perfusion were demonstrated using primary human MPCs and endothelial cells within the same scaffolds [38].

Despite the successful in vivo outcomes of MuSCs in muscle regeneration, the ability to restore muscle function remains in question. For example, MPCs delivered via keratin hydrogels did not contribute to enhanced functional muscle recovery in terms of peak isometric torque production in a rat TA VML injury (Table 2) [43]. Additionally, the co-culturing of myoblasts with microvessel fragments (MVF) in a collagen hydrogel resulted in minimal muscle regeneration in a rat VML model (Table 2) [44]. These studies suggest that many factors, such as finding an optimal microenvironment for muscle cells or the presence of specific paracrine signals, play crucial roles in functional muscle regeneration.

### 4.4. MSC-Based Therapies for Preclinical Treatment of VML

Besides muscle stem and progenitor cells, MSCs have been tested in rodent VML models [45,46]. Recent studies suggest that the pro-regenerative outcomes of these cells are not due mainly to the direct contribution of these cells to de novo myofiber formation, but instead to their paracrine effects, such as immunomodulation, inducing vascularization, mediating with fibrotic response and signaling to endogenous cells. In one study, bone marrow-derived MSCs (BM-MSCs) encapsulated in plasma-rich platelet-derived fibrin microbeads could accelerate muscle regeneration in a rat VML model of the ablated biceps femoris muscle. At 180 days following implantation, animals that did not have microbeads and MSCs showed incomplete repair with scar formation, whereas the presence of microbeads decreased collagen deposition and caused the formation of aligned myofibers. Based on histomorphometric analysis at 30, 60 and 180 days following implantation, MSCs encapsulated in fibrin showed immunomodulatory effects, and sped up the healing process compared to fibrin alone [47].

Similarly, another study using adipose-derived MSCs (ASCs) in collagen hydrogels to treat rat VML demonstrated improved therapeutic outcomes. Laser Doppler perfusion imaging 2 weeks post-injury demonstrated that blood perfusion was doubled in groups with ASC-loaded hydrogels compared to cell devoid hydrogels. The upregulation of endothelial cell markers, such as α-SMA, in these groups also confirmed the angiogenic effect of ASCs. In groups with ASC-loaded hydrogels, anti-inflammatory cytokines such as IL4 and IL10 were increased, and pro-healing macrophage markers such as CD163, CD206 and Arg-1 were upregulated. In addition, myogenic markers, including myogenin and MyoD, were upregulated, and collagen deposition was decreased, which together indicated the immunomodulatory effect of ASCs in attenuating the inflammation and enhancing tissue regeneration [48]. Using a severe rat VML model in which the entire TA and LD muscles were excised, treatment with human ASC-seeded electrospun fibrin scaffolds increased the preservation of scaffold fibers up to four times in the cross-sectional area, compared to that of acellular scaffolds, at 3 months post-implantation. Unlike groups with acellular scaffolds, in ASC-seeded fibers, mature muscle cells that express MyHC were observed. However, it is unclear from this study if ASCs can differentiate and fuse to the host muscle cells, or just improve regeneration through paracrine signals [49].

In addition to ASCs and BM-MSCs, amniotic MSCs demonstrated positive therapeutic effects in VML. Amniotic MSCs were induced using 5-azacytidine, a biochemical induction factor, and delivered to rat TA muscle with VML within a gelatin methacrylate (GelMA) hydrogel. Treatment with MSCs significantly increased neovascularization and functional tissue repair, compared to groups that received just GelMA scaffold without cells or blank groups 2 and 4 weeks post-implantation [50]. Together, these studies suggest that multiple sources of MSCs can have therapeutic effects on the treatment of VML.

Although research on the use of iPSC-derived myogenic studies is still underway, a recent study by Rao et al. evaluated the efficacy of human iPSCs as a source of myogenic cells for muscle regeneration in a mouse model. Human iPSCs within patterned polydimethylsiloxane (PDMS) expressed Pax7 protein and directly contributed to the formation of new myofibers, generating 3D vascularized and contractile muscle constructs [17]. It is likely that the efficacy of such constructs will be tested for treatment of VML in the near future. A summary of recent animal studies of cell-based therapies in VML treatment is presented in Table 2. These promising results in small animal models are a step forward towards finding an effective treatment for VML in humans and the ability to regenerate functional muscle tissue. However, constructing engineered functional tissue in a large volume and at the human scale introduces more challenges in terms of extending ex vivo cells and validating the constructs in larger animal models.

## 5. Conclusions and Future Perspectives

MuSCs and MSCs are the two main cell sources that have been evaluated in preclinical cell therapies in VML rodent models, and have demonstrated the capacity for muscle regeneration. However, the absence of a standardized VML surgical model as well standardized functional measurements of muscle regeneration makes it difficult to compare the results of various studies. Since almost all the present preclinical studies have been performed in small rodent models, the scalability of cell-based therapies for large animal models is a big challenge. MuSCs lose their stemness during in vitro expansion, so the development of a robust in vitro expansion method for MuSCs without losing their potency is necessary. Additionally, more studies are needed in order to understand the molecular mechanisms behind the differentiation and self-renewal behavior of MuSCs. Both murine and human MuSCs have been used for studies in this area, but due to differences between human and murine MuSCs, in vitro culture systems are required to be optimized specifically for human MuSCs. MSCs show superiority in terms of abundance for large defects in humans, but these cells have a poor myogenic differentiation capacity. Human iPSC-derived MuSCs may currently be the only source of infinite autologous cells for scalable therapeutic use, but the tumorigenic potential of iPSCs remains a major concern for clinical translation, and no preclinical studies have evaluated the potential of these cells for VML as yet. For the translation of cell-based therapies into the clinic, developing a delivery vehicle to promote cell survival in in vivo implantation, and obtaining appropriate protocols for cell expansion and storing in vitro, is required. 

The success of cellular therapeutics in treating VML strongly depends not only on the therapeutic cells themselves, but also on microenvironmental cues presented to the cells. Exploiting tissue engineering strategies can improve cell-based therapies for muscle regeneration. However, more work is still needed in order to elucidate the optimized microenvironment for cells capable of regenerating functional muscles, vascularization and innervation. Poor innervation is a major obstacle to achieving the full-force recovery of new muscles, and although the in vitro co-culturing of neurons and muscle cells has been promising, preclinical studies of muscle–nerve constructs are limited. In addition, repairing myotendinous junctions in VML remains an unmet challenge. Further studies are needed that focus on establishing muscle–tendon junctions and the generating of new nerves with new muscles in VML.

Although cell-based therapies to date have shown exciting potential, we believe that it is the combination of cellular therapeutics and tissue engineering approaches that will open up more possibilities for finding a clinical therapy for VML. An improved understanding of the molecular and cellular interactions of muscle regeneration will facilitate the engineering of an optimal microenvironment to guide therapeutic cells toward restoring muscle function after VML. Exploiting emerging technologies, such as on-chip disease models and Slide-Seq, will help in probing the role of different cells present at VML injury sites at the molecular and genomic levels [54,55]. Understanding the complex multicellular and ECM interactions of the inflammatory microenvironment of VML could open new windows for selecting an optimized combination of myogenic and non-myogenic cells in an engineered microenvironment niche in order to induce de novo muscle regeneration.

## Figures and Tables

**Figure 1 bioengineering-07-00097-f001:**
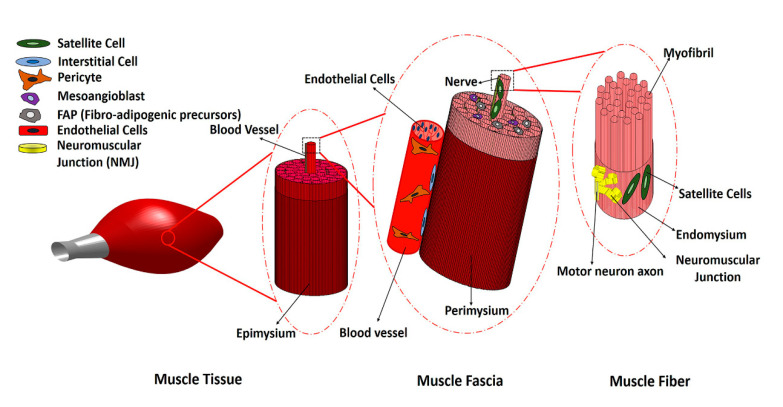
Schematic of skeletal muscle structure and different cell types in skeletal muscle.

**Table 1 bioengineering-07-00097-t001:** Cell Sources for Regeneration of Skeletal Muscle.

Cell Types	Markers	Location	Advantages	Disadvantages	Reference
MuSCs	Pax7^+^, CD56^+^, MyoD^+^	Under basal lamina of muscle fibers.	Critical to native skeletal muscle regeneration. High myogenic potential.	Isolation is invasive and low yield. Loss of self-renewal potential during in vitro expansion. Loss of differentiation potential after in vivo transplantation.	[7,19,20]
Mesenchymal stem cells (MSCs)	CD90^+^, CD44^+^, CD29^+^, CD105^+^, CD13^+^, CD73^+^, CD166^+^, CD45^−^, CD34^−^, CD14^−^	Adipose tissue, bone marrow (BM), umbilical cord (UC).	Abundance of adipose tissue. Ease of isolation from adipose tissue. Low expression of MHC-I and MHC-II Immunomodulatory effect.	Invasive isolation for BM-MSCs. Poor myogenic differentiation capacity.	[7,19]
Myo-endothelial cells	CD34^+^, CD144^+^, CD56^+^, CD31^+^, CD45^−^	Periphery of myofibers close to blood vessels.	Have both angiogenic and myogenic capacity.	Laborious isolation and purification process. Limited literature on their role in skeletal muscle regeneration.	[21]
Mesoangioblasts	CD34^+^, Sca-1^+^, CD31^+^, c-Kit^+^, CD45^−^	Walls of microvessels.	High proliferative capacity in vitro. Multipotent cells with potential to differentiate into skeletal muscle	Invasive isolation procedure. Lack of studies for VML treatment.	[22]
Pericytes	CD146^+^, NG2^+^, ALP^+^, PDGFR-β^+^	Periphery of capillaries and microvessels.	Pericyte myogenesis naturally occurs during development and regeneration of muscle. High muscle differentiation potential. Lack of MHCII expression.	Limited literature on their potential in skeletal muscle regeneration and VML.	[7,19]
CD133^+^ progenitor cells	CD133^+^, CD34^+/−^, CD90^+/−^, CD146^+^	Periphery of myofibers close to blood vessels.	Availability and ease of purification from peripheral blood Myogenic and angiogenic capacity.	Reduction of myogenic potential following in vitro culture.	[19,23]
Induced pluripotent stem cells (iPSCs)	Oct4^+^, Sox2^+^, KLF4^+^, and c-Myc^+^	All tissues, mainly skin.	Unlimited self-renewal in vitro. Patient-derived autologous cells. Myogenic differentiation capacity.	Risk of tumorigenicity and genetic instability.	[7,19]
Embryonic stem cells (ESCs)	Oct4^+^, Sox2^+^, KLF4^+^, and c-Myc^+^	Inner cell mass of blastocyst.	Unlimited self-renewal in vitro. Myogenic differentiation capacity	Ethical concerns Inefficient isolation process. Risk of tumorigenicity. Risk associated with immune response.	[7]
Muscle side population cells (SPs)	CD45^−^, c-Kit^−^, Sca1^+^, ABCG2^+^, Pax7^−^, Myf5^−^, Desmin^−^	Interstitial space of skeletal muscle.	Myogenic differentiation capacity in vivo.	Low availability Lack of specific phenotypic markers. Poor myogenic differentiation in vitro. Limited literature on their potential for skeletal muscle regeneration and VML.	[20,24]

Abbreviations: Stem cells antigen -1 (Sca-1), Alkaline phosphatase (ALP), Platelet-derived growth factor receptor beta (PDGFR-β), Neural/glial antigen 2 (NG 2), ATP binding cassette subfamily G member 2 (ABCG-2), Octamer-binding transcription factor 4 (Oct-4), SRY (sex determining region Y)-box 2 (Sox-2), Kruppel-like factor 4 (KLF4), Myogenic factor 5 (Myf5).

**Table 2 bioengineering-07-00097-t002:** Animal studies of cell-based therapies in VML treatment.

Cell Type	In Vitro Findings	Animal Model	Delivery Technique	In Vivo Findings	Reference
MPCs ASCs	-	Murine LD muscle (50% defect).	Cells seeded in Bladder acellular matrix (BAM) scaffolds.	Histological and immunohistochemical analysis shows ADSCs could create regenerated muscle comparable to MPCs seeded scaffolds, but mainly through participation in vascularization.	[45]
Human UC-MSCs	-	Rat TA muscle (20% defect).	Placing cells in aggregate in the muscle defect with and without decellularized porcine heart ECM powder.	Histological analysis and mechanical function evaluation show MSCs and decellularized ECM have a synergistic effect on promoting skeletal muscle regeneration.	[46]
Combination of MuSCs, ECs, FAPs, hematopoietic cells, fibroblast like cells	Bioluminescence imaging (BLI) measurements demonstrated viability was significantly enhanced in the presence of support cells. Ex vivo force measurement shows force recovery reaches up to 90% of the uninjured muscle.	Murine TA muscle (40% defect).	Decellularized murine TA ECM-based hydrogel.	The combination of cells with scaffolds could generate functional vascularized muscle tissue in VML models; however, innervation and muscle force are not sufficient, yet could be enhanced by exercise.	[42]
Human skeletal muscle cells (hSKMCs)	Printed 3D cell constructs demonstrate high cell viability (>90%), differentiation, myotube formation and contractility.	Rat TA muscle (40% defect)	Cell-laden muscle decellularized ECM (mdECM) bioink.	Pre-vascularized 3D cell printed muscle constructs improve muscle regeneration, vascularization and innervation, as well as 85% of functional recovery.	[51]
ASCs	ASCs proliferate and align on fibers with acceptable cell viability, but do not fully express myotube characterization and myogenesis fails after 2 months in vitro.	Murine TA and extensor digitorum longus (EDL) removal.	Cells-seeded electrospun fibrin scaffold.	ASCs combined with electrospun fibrin microfibers demonstrate more tissue regeneration in vivo compared with acellular fibers, but limited expression of myogenic markers in ASCs is observed.	[49]
Human MPCs	-	Murine TA muscle.	Poly-lactic-glycolic acid (PLGA) 3D scaffold.	Scaffolds increase the viability of cells in vivo and regeneration of muscle is enhanced following 1 and 4 week implantation compared to direct cell injection.	[52]
Rat Bone-marrow MSCs	-	Rat biceps femoris resection size: 8 × 4 × 4 mm^3^.	Fibrin-based microbeads.	Fibrin microbeads with and without MSCs accelerate muscle regeneration and prevent scar formation; MSCs shorten the regeneration period. Sham group has in incomplete repair and fibrotic scar formation.	[47]
Rat ASCs	-	Rat TA muscle resection size: 10 × 5 × 3 mm^3^.	Type I hydrogel.	ASCs encapsulated in hydrogel reduced inflammation and collagen deposition and accelerated muscle regeneration and angiogenesis compared with the hydrogel group.	[48]
Human ASCs	Viability and growth of ASCs on electrospun fibers were assessed by Live/Dead and PicoGreen assays for up to 21 days. After 2 months in culture, both induced and uninduced ASCs formed elongated and aligned fibers on electrospun fibers and expressed high levels of desmin, but they expressed low and non-nuclear Myogenin and could not fully recapitulate myotube formation.	Removal of TA and EDL muscles from the anterior tibial compartment in immunodeficient mice.	Electrospun fibrin hydrogel microfiber bundles.	ASC-seeded fibers exhibited up to four times higher volume retention than acellular fibers and lower levels of fibrosis. Unlike acellular scaffolds, ASC-seeded scaffolds showed mature muscle cells.	[49]
Human amniotic MSCs	Results of Live/Dead test and immunofluorescence staining of desmin and MyoD showed that the cell viability and induction of the myogenic differentiation of hAMCs by 5-Aza was not affected by GelMA gel.	Sprague Dawley (SD) rats 5 mm diameter muscle defect in TA muscle using a hole punch.	GelMA gel.	Results showed 5-Aza induced cells in GelMA reduced the scar formation and increased the vascularization 2 weeks and 4 weeks post-implantation compared to blank and GelMA groups.	[50]
Microvessel fragment (MVF) construct with myoblasts (MVF + Myoblasts)	Live/Dead assay demonstrates high viability of microvessels and seeded myoblasts and immunofluorescent staining shows microvessel networks increase more in MVF-Myoblast constructs than in MVF-only constructs.	12 mm biopsy punch in biceps femoris muscle of Sprague Dawley rats.	Collagen hydrogel.	MVF-Myoblast constructs did not show muscle regeneration at both 2 weeks and 8 weeks post-implantation.	[44]
Rat MPCs		Adult female Lewis rats 20% TA muscle.	Keratin hydrogel.		[43]
Mouse MPCs		Female C57/BL6 Mouse 50% LD muscle.	Keratin hydrogel.		[53]
Newborn mice MuSCs	-	Three month old immunodeficient NSG mice TA muscle 4 × 2 × 2 mm^3^ partial thickness wedge resection.	Fibrin hydrogel	Transplanted MuSCs in fibrin contribute to forming new fibers and new vessels and increase muscle mass as well as reduce fibrotic response.	[41]
Human MPCs and human microvascular endothelial cells	Human MPCs expressed Pax7 protein and were aligned along the direction of the scaffold nanofibers.	20% TA muscle ablation in NOD SCID male mice.	Nanofibrillar collagen scaffold.	Vascular perfusion and donor-derived human myofiber density increased in endothelialized human skeletal muscle formed from aligned scaffolds compared to randomly-oriented scaffolds.	[38]

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
