# Peer review of "Pre-Clinical Cell Therapeutic Approaches for Repair of Volumetric Muscle Loss"

_bioengineering, 2020, doi:10.3390/bioengineering7030097_

Round 1

Reviewer 1 Report

Thank you for the opportunity to read your manuscript. I understand that the aim of your work is to provide a condensed overview of the current knowledge on cell-based therapies for muscle regeneration and to point out possible future trends in this field. I appreciate that you first described the process of endogenous muscle regeneration and discussed its natural limitations. I also appreciate that you then gave an overview of different myogenic and non-myogenic cell types that are involved in the process of muscle healing and/or could support this process as a therapy. I also appreciate that in the second part of your manuscript you refer in more detail to corresponding experimental studies that have investigated the principal regenerative effects of different cell-based approaches in preclinical models.

Although your manuscript is generally well structured, it is somewhat "weak" in its scope and, above all, a critical discussion of the presented data is missing, e.g. on the different cell sources for muscle regeneration. You gave a good and very condensed overview of the most important key points of the general topic,but  your work does not go far beyond the already published reviews or is described in much more detail in them (e.g. PMID: 30843380; PMID: 28575733, PMID: 25271446; PMID: 29534011, or https://doi.org/10.1021/acsbiomaterials.7b00005). The critical examination of existing and frequently used animal models for VML in relation to clinical reality is not presented in sufficient detail, nor is an overview of the first clinical trials investigating such cell-based therapies in humans. Therefore, I regret to say that your review and in particular your chapter "Conclusion and Future Perspectives" contributes little to the current understanding in this field or largely presents ideas that have already been described in more detail in other papers.

Therefore, I would suggest that you emphasize the differences between your work and existing reviews more clearly in the introduction. Your focus on VML is very useful in this respect, but you should present the current (clinical) therapeutic options more comprehensively (please refer e.g. to PMID: 27825146, 29843673 or 30843380). In addition, I recommend (as mentioned above) to present and to compare the most important animal models in more detail and critically discuss their advantages and disadvantages with regard to mimicking human / patients. The sections on myogenic and non-myogenic cell types are good in terms of content, but require linguistic revision. A detailed section on suitable biomaterials for providing regenerative cells or improving their function in vivo would also be very useful. Finally, you should try to describe/mention clinical studies testing such cell-based therapies for muscle regeneration (please refer to e.g. PMID: 31578930 or the above mentioned). I am aware that there is not much more on this topic, but it is important to discuss why not (what is missing, what is needed? What are current general limitations?), or what is the most promising approach that could soon enter clinical evaluation - and why this one and not the others?

In summary, I regret to say that I must reject your work as it stands. I am aware that you are very disappointed with this assessment, but I would like to say that my recommandation is not necessarily a reflection on the general quality of your work, but in my opinion the novelty is too limited to recommend publication or major revision.

Author Response

  • Although your manuscript is generally well structured, it is somewhat "weak" in its scope

We agree with the reviewer that we need to highlight the scope of the manuscript, which is the status of cell-based therapies for VML treatment. To address this issue, we added a paragraph in the introduction section, page 1, lines 27-37, to explain more about VML and the current treatments. We’ve also added a few sentences at the end of introduction (page 2, lines 60-63), explicitly describing the scope of the manuscript.

  • A critical discussion of the presented data is missing, e.g. on the different cell sources for muscle regeneration.

Different myogenic cell sources have been introduced individually and their advantages and limitations have been discussed. To classify and easily compare these cell sources, critical characteristics of them along with their pros and cons have been presented in table 1. We added a new paragraph in page 3, lines 119-126 to critically discuss the key points..

  • You gave a good and very condensed overview of the most important key points of the general topic, but  your work does not go far beyond the already published reviews or is described in much more detail in them (e.g. PMID: 30843380; PMID: 28575733, PMID: 25271446; PMID: 29534011, or https://doi.org/10.1021/acsbiomaterials.7b00005).

We agree that our paper contains many of information in other reviews in this field; however, none of these papers have focused on cell-based treatments specifically for VML. Our aim is to focus on VML disease and study the current status of cell-based strategies for VML.

  • The critical examination of existing and frequently used animal models for VML in relation to clinical reality is not presented in sufficient detail, nor is an overview of the first clinical trials investigating such cell-based therapies in humans. Therefore, I regret to say that your review and in particular your chapter "Conclusion and Future Perspectives" contributes little to the current understanding in this field or largely presents ideas that have already been described in more detail in other papers. Therefore, I would suggest that you emphasize the differences between your work and existing reviews more clearly in the introduction. Your focus on VML is very useful in this respect, but you should present the current (clinical) therapeutic options more comprehensively (please refer e.g. to PMID: 27825146, 29843673 or 30843380).

Thank you for your comments. We added two paragraphs in the introduction section to describe more about VML and current clinical options and studies and their limitations and why we need to know about cellular therapeutics for VML repair. (Page 2, Lines 41-57)

  • In addition, I recommend (as mentioned above) to present and to compare the most important animal models in more detail and critically discuss their advantages and disadvantages with regard to mimicking human / patients. The sections on myogenic and non-myogenic cell types are good in terms of content, but require linguistic revision.

We added one sub-section for animal models (Page 7, Lines 172-193) and discussed the suggested content of the reviewer. The paper has been revised.   

  • A detailed section on suitable biomaterials for providing regenerative cells or improving their function in vivo would also be very useful.

We added one section in preclinical studies for biomaterials (Page 7, Lines 195-210).

  • Finally, you should try to describe/mention clinical studies testing such cell-based therapies for muscle regeneration (please refer to e.g. PMID: 31578930 or the above mentioned). I am aware that there is not much more on this topic, but it is important to discuss why not (what is missing, what is needed? What are current general limitations?), or what is the most promising approach that could soon enter clinical evaluation - and why this one and not the others?

Thank you for your suggestions. We added requested information in the following paragraphs (Page 2, Lines 41-57)

Reviewer 2 Report

The review neatly discusses the current knowledge about different cell types and their usage in cell therapeutic approaches for the treatment of Volumetric Muscle Loss. Well written, it nonetheless solely displays a restrained summary of the existing literature.

Even though MSCs, MuSCs, IPS and FAPs are described as potential players, the review lacks scientific profundity as in critical evaluation and innovative approaches. There is no obvious translational attempt in the analysis and assessment of the references and examples used.

Further, the authors do not have a critical view on the usage of MSCs from different sources and the associated challenges.

A clinical approach, such as the usage of muscle flaps, is only briefly mentioned once in the introduction. Surely, there is more information on techniques available, which are being used in a clinical set-up. In this context, one would expect a discussion on pursuing complications in the standard of care treatment. The review offers hardly any insight into current clinical trial endeavors in the context of VML.

Discussing innovative cell therapeutic approaches also calls for critically considering applications methods and materials, particularly with regard to a potential clinical implementation.

In line 21 the authors promise a “focus on preclinical testing and future trends in this field “. Even though some different animal models are mentioned, again the authors do not state, which animal model cited reflects or displays in detail the corresponding clinical issues. A more revealing analysis and critical evaluation of the animal models available with reference to the human clinical situation would be desirable.

The title of the manuscript is slightly misleading. “Cellular Therapies for Skeletal Muscle Regeneration” includes more aspects of regeneration than touched by the current version. A more precise title, such as “Pre-clinical cell therapeutic approaches addressing volumetric muscle loss” would be more suitable.

Author Response

  1. Even though MSCs, MuSCs, IPS and FAPs are described as potential players, the review lacks scientific profundity as in critical evaluation and innovative approaches. There is no obvious translational attempt in the analysis and assessment of the references and examples used.

To address the reviewer’s concern, we added multiple paragraphs to resolve the lack of critical evaluation (Page 3, Lines 119-126; Page 6. 149-153; Page 15, Lines 327-329; Page 15, Lines 346-351)

  1. Further, the authors do not have a critical view on the usage of MSCs from different sources and the associated challenges.

We added a few sentences to address this comment (Page 3, Lines 92-96, 101-103 and 119 to 126).

  1. A clinical approach, such as the usage of muscle flaps, is only briefly mentioned once in the introduction. Surely, there is more information on techniques available, which are being used in a clinical set-up. In this context, one would expect a discussion on pursuing complications in the standard of care treatment. The review offers hardly any insight into current clinical trial endeavors in the context of VML. Discussing innovative cell therapeutic approaches also calls for critically considering applications methods and materials, particularly with regard to a potential clinical implementation.

We agree with the reviewer. Hence, we added two paragraphs to discuss the current clinical options for VML treatment and their limitations. (Page 1, Lines 27-37 and Page 2, Lines 41-57).

  1. In line 21 the authors promise a “focus on preclinical testing and future trends in this field “. Even though some different animal models are mentioned, again the authors do not state, which animal model cited reflects or displays in detail the corresponding clinical issues. A more revealing analysis and critical evaluation of the animal models available with reference to the human clinical situation would be desirable.

We added one sub-section for animal models (Page 7, Lines 172-193) and discussed the suggested content of the reviewer.

  1. The title of the manuscript is slightly misleading. “Cellular Therapies for Skeletal Muscle Regeneration” includes more aspects of regeneration than touched by the current version. A more precise title, such as “Pre-clinical cell therapeutic approaches addressing volumetric muscle loss” would be more suitable.

We changed the title.

Reviewer 3 Report

Shayan et al provide a review of stem cell strategies for the challenging and complex VML injury in skeletal muscle. There is a clinical need for tissue engineering therapies in VML, and since stem cells are a promising part of that future, a review on the subject is useful and timely. The primary concern is that the main focus of this review is not immediately clear. Further, the review is brief, and certain sections are a little too general. Some do not include critical information about a topic or how it specifically relates to VML. While certain topics or details may be out of the scope of the review, that scope is not always clear.  

  1. From the abstract and introduction, VML seems to be a focal point of the review. It would be helpful to provide a more detailed consensus description on the definition of a VML injury at the beginning of the review to orient the reader. The nuances of the definition can still be expanded later, but at least a working definition would be very helpful.
  2. MSCs are very briefly mentioned, and this section only seems to comment that additional progress is required for use in muscle regeneration. It would be helpful to more specifically discuss the myogenic potential of MSCs (or lack thereof in most cases), thus justifying the limitations.
  3. In a large proportion of contexts, FAP activity is implicated in adipose and fibrotic tissue accumulation within muscle. The FAP description should be balanced to include these prominent negative roles in muscle repair.
  4. The earliest formal VML descriptions (Grogan et al) were driven by loss of muscle function. Though preclinical models use mass lost as a necessary means to standardized, the functional aspects could use a good deal more discussion..
  5. What are the unique challenges for VML that are not captured in traditional muscle repair models (e.g. cardiotoxin)? These complex differences are what make VML injuries uniquely difficult to treat, and provide a true need for tissue engineering approaches. (stem cells alone may make sense for dystrophy, but are unlikely to be successful for VML without a scaffold)
  6. Minor typo: ArgI should be Arg1.
  7. The conclusion mentions a lack of consistency in functional measurements, but functional testing is not mentioned elsewhere in the manuscript. Please discuss. (and a similar comment for innervation)

Author Response

1. From the abstract and introduction, VML seems to be a focal point of the review. It would be helpful to provide a more detailed consensus description on the definition of a VML injury at the beginning of the review to orient the reader. The nuances of the definition can still be expanded later, but at least a working definition would be very helpful.

We have added a paragraph in introduction to address this issue (Page 1, Lines 27-37).

2. MSCs are very briefly mentioned, and this section only seems to comment that additional progress is required for use in muscle regeneration. It would be helpful to more specifically discuss the myogenic potential of MSCs (or lack thereof in most cases), thus justifying the limitations.

We have added the suggested content to the manuscript (Page 3, Lines, 101-103).

3. In a large proportion of contexts, FAP activity is implicated in adipose and fibrotic tissue accumulation within muscle. The FAP description should be balanced to include these prominent negative roles in muscle repair.

Thank you for your comment. We totally agree with the reviewer and have added a few sentences to explain positive and negative roles of FAPs in muscle regeneration and VML (Page 6, Lines 141-143).

4. The earliest formal VML descriptions (Grogan et al) were driven by loss of muscle function. Though preclinical models use mass lost as a necessary means to standardized, the functional aspects could use a good deal more discussion.

Thank you! We agree and we mentioned Anderson work (Anderson, Shannon E., et al. "Determination of a critical size threshold for volumetric muscle loss in the mouse quadriceps." Tissue Engineering Part C: Methods 25.2 (2019): 59-70.)  which investigated the critical defect size for VML model based on the fibrotic response and lack of neuromotor and myofiber regeneration.

5. What are the unique challenges for VML that are not captured in traditional muscle repair models (e.g. cardiotoxin)? These complex differences are what make VML injuries uniquely difficult to treat, and provide a true need for tissue engineering approaches. (stem cells alone may make sense for dystrophy, but are unlikely to be successful for VML without a scaffold)

Thank you for pointing this out. We have added a paragraph in the introduction explaining why treating VML is challenge (Page 2, Lines 41-49)

6. Minor typo: ArgI should be Arg1.

Corrected (Page 7, Line 275).

7. The conclusion mentions a lack of consistency in functional measurements, but functional testing is not mentioned elsewhere in the manuscript. Please discuss. (and a similar comment for innervation)

Lack of consistency in measurement methods used for assessing therapeutic strategies for VML treatment is a limitation for comparing the efficacy between different treatments. For example, many studies have just assessed myofiber formation using histological analysis; however they did not analyze the force restoration of the muscle or innervations. We added a few sentences to clarify this key point (Page 8. Lines 222-224).

Round 2

Reviewer 1 Report

Thank you for going into the reviewers' comments in such detail. From my point of view, your manuscript has improved substantially.

I would like to point out minor inaccuracies that you should improve in later steps of the editing process:

Line 92: "models [12] - blank space is missing.

Line 100-103: "many studies have shown low engraftment of MSCs and lack of tissue-specific differentiation. It is not clear that MSCs are beneficial in muscle regeneration just by secreting immunomodulatory and paracrine signals or by myogenic differentiation." For this you should give one or two corresponding references (preferably corresponding reviews).

Reviewer 2 Report

You have well adressed my previous concerns by adding relevant paragraphs and data and I believe the manuscript is suitable for publication in the present form.